# VHSMarker and the CCK Dataset: A Benchmark for Automated Vertebral Heart Score Estimation in Canine Radiographs

## Abstract

We present VHSMarker, a web-based annotation tool that enables rapid and standardized labeling of six cardiac key points in canine thoracic radiographs. VHSMarker reduces annotation time to 10–12 seconds per image while supporting real-time vertebral heart score (VHS) calculation, model-assisted prediction, and quality control. Using this tool, we constructed the Canine Cardiac Key Point (CCK) Dataset, a large-scale benchmark of 21,465 annotated radiographs from 12,385 dogs across 144 breeds and additional mixed breed cases, making it the largest curated resource for canine cardiac analysis to date. To demonstrate the utility of this dataset, we introduce MambaVHS, a baseline model that integrates Mamba blocks for long-range sequence modeling with convolutional layers for local spatial precision. MambaVHS achieves 91.8% test accuracy, surpassing 13 strong baselines including ConvNeXt and EfficientNetB7, and establishes state-space modeling as a promising direction for veterinary imaging. Together, the tool, dataset, and baseline model provide the first reproducible benchmark for automated VHS estimation and a foundation for future research in veterinary cardiology. The source code and dataset are available on our project website: https://anonymousgenai.github.io/vhsmarker.

## 1 Introduction

Canine cardiomegaly, characterized by pathological heart enlargement, is a critical condition that can significantly impact the health and longevity of dogs if left undiagnosed or untreated. Early and accurate detection is essential for effective intervention, yet traditional methods for Vertebral Heart Score (VHS) measurement from thoracic radiographs remain highly subjective, labor-intensive, and prone to inter-observer variability (Bappah et al., 2021; Burti et al., 2020). These manual processes require precise anatomical landmark identification and measurement, demanding significant expertise and time, which limits their scalability in clinical practice (Rungpupradit & Sutthigran, 2020; Dumortier et al., 2022). Recent advances in deep learning, including convolutional neural networks (CNNs) (He et al., 2016; Huang et al., 2017), transformer-based architectures (Dosovitskiy et al., 2020), and state-space models such as MambaVision (Hatamizadeh & Kautz, 2024), have demonstrated exceptional potential for medical image analysis, often outperforming conventional rule-based methods in both accuracy and efficiency. These approaches can capture long-range dependencies and complex spatial relationships, making them suitable for challenging imaging tasks. However, their application to veterinary diagnostics remains limited due to the scarcity of large, high-quality labeled datasets, the diversity of canine anatomies, and the need for interpretable and clinically reliable predictions (Jeong & Sung, 2022; Zhang et al., 2021).

To address these challenges, we introduce a comprehensive framework that unifies an annotation tool, a large-scale dataset, and a baseline model for canine cardiac assessment. Unlike prior veterinary cardiology studies that relied on small datasets or isolated methods (Bappah et al., 2021; Burti et al., 2020; Jeong & Sung, 2022; Zhang et al., 2021), our contribution establishes the first standardized benchmark for automated VHS estimation, providing the community with both resources and strong baselines.

Our framework makes three key contributions. (i) We present a clinician-oriented web tool that reduces annotation time from over a minute to about 10–12 seconds per image, while supporting real-time keypoint placement, automated VHS calculation, built-in quality checks, and seamless data export, enabling scalable and accurate dataset creation with minimal user error. (ii) We introduce the Canine Cardiac Keypoint (CCK) Dataset, a carefully curated collection of over 21k radiographs annotated with six cardiac keypoints, offering a standardized benchmark that captures diverse anatomical variations and clinical conditions for training and evaluation. (iii) We develop MambaVHS, a hierarchical baseline model that combines Mamba blocks for efficient long-range sequence modeling with convolutional layers for local spatial precision, achieving robust and accurate VHS prediction that surpasses existing baselines.

Together, the tool, dataset, and model form an end-to-end pipeline for automated cardiomegaly assessment. This framework significantly reduces annotation burden, provides a reproducible benchmark, and demonstrates through MambaVHS that state-space modeling is a promising direction for veterinary imaging tasks.

## 2 RELATED WORK

The diagnosis of canine cardiomegaly has traditionally relied on the Vertebral Heart Score (VHS), which measures cardiac dimensions relative to thoracic vertebrae (Bappah et al., 2021; Rungpupradit & Sutthigran, 2020; Buchanan & Bücheler, 1995). While widely used in veterinary practice, VHS measurement suffers from inter-observer variability and time-consuming manual processes (Litster & Buchanan, 2005; Lam et al., 2001). Efforts to improve consistency include standardized protocols (Jeong & Sung, 2022) and computational methods (Rungpupradit & Sutthigran, 2020), yet these still depend on manual initialization.

Automated tools for VHS have been explored. Li, Zhang introduced a MATLAB-based system requiring manual adjustment (Li & Zhang, 2024), while Oh, Lee, Go, Lee, and Jeong (Oh et al., 2024) proposed a semi-automated segmentation pipeline that reduces manual oversight by leveraging few-shot learning. Fully automated solutions are more advanced in human cardiology (Alsharqi et al., 2018; Zhang et al., 2021), but remain difficult to adapt to veterinary settings due to anatomical differences and limited data.

Deep learning has transformed medical imaging, with CNNs excelling at segmentation and disease detection (Huang et al., 2017; Ronneberger et al., 2015; Dumortier et al., 2022; Wang et al., 2017). In veterinary applications, CNNs have been applied to canine cardiomegaly (Burti et al., 2020), feline pulmonary disease (Dumortier et al., 2022), and bovine teat-end analysis (Zhang et al., 2022), though such studies are constrained by small datasets and lack of standardized benchmarks (Litjens et al., 2017; Jeong & Sung, 2022). More recent advances include transformers (Dosovitskiy et al., 2020; Liu et al., 2022a; Wu et al., 2021), which capture long-range dependencies, and Mamba-based architectures (Hatamizadeh & Kautz, 2024; Gu & Dao, 2023), which achieve linear-time sequence modeling. These capabilities are especially relevant for VHS, where distant landmarks must be jointly modeled. Zhang et al. (2025) further explored diffusion-based augmentation for canine cardiomegaly, but focused on data generation rather than standardized landmark localization.

Overall, veterinary cardiology research remains limited by scarce annotated datasets, reliance on manual or semi-automated tools (Li & Zhang, 2024; Dumortier et al., 2022; Zhang & Davison, 2021), and the absence of reproducible evaluation pipelines. Our work addresses these gaps by introducing the first comprehensive benchmark: a scalable web-based annotation tool (VHSMarker), the large-scale CCK Dataset with standardized keypoints, and MambaVHS as a strong baseline model, enabling reproducible evaluation and exploration of state-space architectures for veterinary imaging.

## 3 DATASET

We introduce the Canine Cardiac Keypoint (CCK) dataset, a benchmark for vertebral heart score (VHS) estimation and cardiac keypoint detection in veterinary cardiology. It provides large-scale radiographs annotated with six cardiac keypoints to support reproducible model training and evaluation. Below we outline the collection process, preprocessing, demographics, and final composition.

Table 1: Demographic statistics including sex, age, and the top 10 breeds. Complete breed distribution is in Appendix A.3.

| Sex Distribution | | Age Distribution (years) | | Top 10 Breeds | | | |
|---|---|---|---|---|---|---|---|
| Category | Count | Age Group | Count | Breed | Count | Breed | Count |
| Female | 7 941 | 0–5 | 2 961 | Mixed Dog | 1 256 | Boxer | 79 |
| Male | 4 395 | 6–11 | 6 272 | Labrador Retriever | 479 | Shih Tzu | 77 |
| Unknown | 49 | 12–17 | 2 827 | Golden Retriever | 191 | Yorkshire Terrier | 77 |
| | | 18–30 | 86 | German Shepherd | 164 | Border Collie | 57 |
| | | Unknown | 239 | Chihuahua | 100 | Beagle Hound | 56 |
| **Total** | **12 385** | **Total** | **12 385** | Top-10 breeds subtotal: **2 536**; Overall total: **12 385** | | | |

**Data and Preprocessing.**   We collected 36,264 canine thoracic radiographs from multiple hospitals under data-sharing agreements and de-identified them.  After quality control, we retained 21,465 *lateral* views (left/right) from 12,385 dogs, excluding dorsoventral/ventrodorsal projections as unsuitable for VHS. Standardized preprocessing removed distorted, overexposed, incomplete, or motion-affected scans, and annotators used VHSMarker's validity flag to exclude clinically irrelevant cases.  Institutional identities remain undisclosed for privacy; the final cohort spans diverse clinical populations and contains only diagnostically sound lateral radiographs for key-point annotation and VHS estimation.

**Demographic Information.**   To assess dataset diversity and representativeness, we report aggregate demographic statistics. The dataset spans 144 distinct dog breeds and a small set of unidentified samples, reflecting broad coverage of anatomical and clinical variability. Although institutional and geographic details remain anonymized for privacy, the CCK Dataset was collected across multiple veterinary hospitals, ensuring diversity in patient populations and imaging practices.  This broad sampling helps mitigate concerns about representativeness and supports the dataset's generalizability to real-world veterinary scenarios. Table 1 summarizes sex distribution, age groups, and the most frequent breeds, while the complete breed distribution (146 entries) is provided in the appendix A.3.

**Final Dataset Composition.**   The Canine Cardiac Keypoint (CCK) dataset comprises 21 465 lateral thoracic radiographs, each annotated with six cardiac keypoints using the VHSMarker tool (Table 2).  The split is performed at the *patient level* (i.e., by dog) to prevent data leakage, ensuring that radiographs from the same individual do not appear across training, validation, and test sets. This design supports robust evaluation and generalization across diverse clinical cases. Combined with precise annotations and integrated quality control, the dataset establishes a reproducible benchmark for vertebral heart score estimation and canine cardiology research.

Table 2: Dataset distribution.

| Split | Images |
|---|---|
| Training | 15026 |
| Validation | 2155 |
| Testing | 4275 |
| **Total** | **21465** |

## 4 METHODS

This section introduces the two main components of our framework.  VHSMarker is a clinician-friendly tool for rapid, standardized keypoint labeling with automated VHS computation, enabling creation of the large-scale CCK Dataset. MambaVHS is a baseline model that integrates convolutional layers with Mamba blocks for precise and efficient VHS estimation.

### 4.1 KEY POINT ANNOTATION TOOL

VHSMarker is a lightweight web-based system for canine cardiac key point annotation. The front end, built with HTML5, JavaScript, and a Canvas interface, enables intuitive point placement and real-time visualization. A Flask back end manages GPU-accelerated inference and asynchronous updates, ensuring low-latency interaction for both expert and non-expert users.

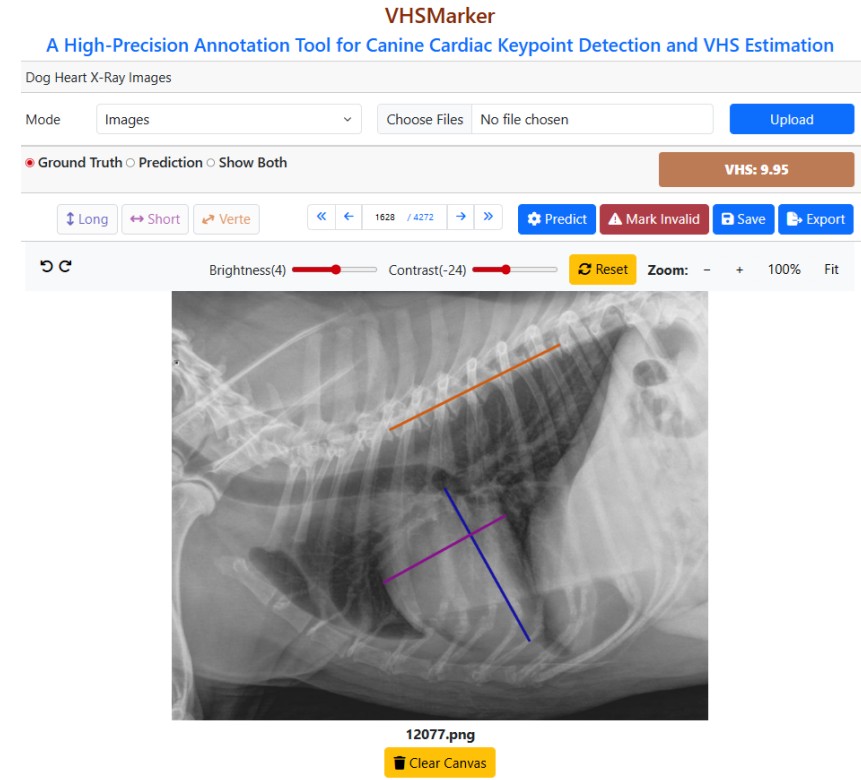

Figure 1: Overview of the VHSMarker interface, including key point placement and real-time VHS calculation.

**Annotation Features.** The VHSMarker interface is designed to balance precision and usability for annotators with varying levels of expertise. It supports zooming, panning, and window-level adjustments (brightness and contrast) to allow close inspection of fine anatomical structures, while undo/redo functions provide stepwise correction without disrupting the workflow. Problematic radiographs can be flagged as invalid, ensuring that only diagnostically reliable samples contribute to the dataset. All annotations, including keypoint coordinates, metadata, and validity flags, are automatically stored in .mat format for seamless downstream integration. To support flexible workflows, three annotation modes are provided: *Ground Truth* for manual labeling, *Prediction* for automated keypoint generation by the MambaVHS model, and *Show Both* for side-by-side comparison and correction (see Appendix A.1). This design enables efficient correction of automated outputs while preserving transparency between human and model contributions. A video demonstration is available on the project website to illustrate the tool's usage in practice.

**Real-Time Inference and VHS Computation.** Upon image upload, the MambaVHS model generates key point predictions, which are overlaid on the canvas. Pixel coordinates $(x_i, y_i)$ are normalized to dimensionless form:

$$\tilde{x}_i = \frac{x_i \cdot \frac{W'}{W}}{H'}, \quad \tilde{y}_i = \frac{y_i \cdot \frac{H'}{H}}{H'}, \tag{1}$$

where $W, H$ are original dimensions and $W', H'$ the target size. The vertebral heart score (VHS) is then computed as:

$$\text{VHS} = 6 \times \frac{(AB + CD)}{EF}, \tag{2}$$

with $AB$ the long axis, $CD$ the short axis, and $EF$ the vertebral reference length. This ensures consistent VHS estimation across variable-resolution images.

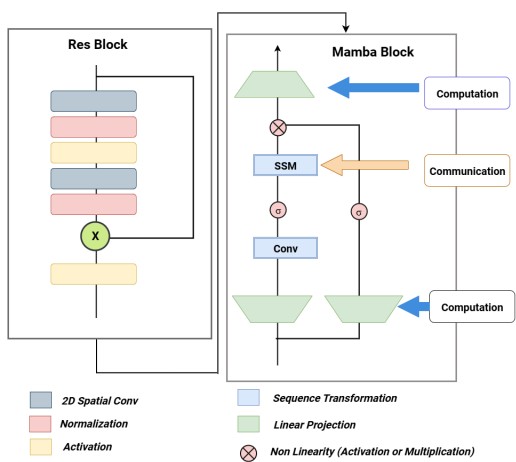

Figure 2: Architecture of the MambaVHS model. It consists of a stem, four MambaStages, and a regression head, combining residual blocks, Mamba SSMs, and SE layers for keypoint prediction.

## 4.2 MAMBAVHS MODEL ARCHITECTURE

The MambaVHS model is a hierarchical deep learning framework designed for precise localization of six cardiac key points in canine thoracic radiographs. Unlike standard CNNs, which primarily capture local context, or Transformers, which model long-range dependencies at quadratic cost, MambaVHS leverages state-space models (SSMs) to capture global anatomical relationships with linear complexity.

The architecture integrates convolutional layers for spatial precision, Mamba blocks for efficient long-range reasoning, and SE layers for adaptive channel recalibration. Training is further guided by the proposed VHSAwareLoss, which embeds clinically meaningful thresholds for vertebral heart score prediction, ensuring that optimization is directly aligned with veterinary diagnostic standards.

Figure 3: Residual (left) and Mamba (right) blocks form each MambaStage: Residual captures local spatial patterns, while Mamba models long-range dependencies.

**MambaStem.** The stem block reduces spatial resolution while expanding feature depth, producing a compact yet expressive representation. It consists of two convolutional layers (stride 2 and stride 1), each followed by batch normalization and SiLU activation:

$$\phi(\mathbf{X}) = \text{SiLU}(\text{BN}(\text{Conv2D}(\mathbf{X}))), \quad (3)$$

$$\mathbf{F}_0 = \phi(\mathbf{X}), \quad (4)$$

where $\mathbf{X}$ is the input radiograph. This operation encodes texture and contour information critical for cardiac structure analysis while reducing computation in later stages.

**MambaStages.** MambaStages refine features via downsampling, residual learning, Mamba-based sequence modeling, and SE recalibration, enabling the model to capture both local anatomical cues and global dependencies for accurate VHS estimation.

*Downsampling.* Spatial resolution is reduced by applying the convolution–BN–SiLU operator from Eq. 3:

$$\mathbf{F}_d = \phi(\mathbf{X}), \quad (5)$$

where $\mathbf{X} \in \mathbb{R}^{B \times C \times H \times W}$ is the input feature map, and $\mathbf{F}_d$ has higher channel depth with reduced spatial size.

*Residual Block.* Local features are captured using two stacked convolutions with a residual skip connection:

$$\mathbf{F}_r = \text{SiLU}(\mathbf{F}_d + \phi(\phi(\mathbf{F}_d))), \quad (6)$$

which preserves fine structural details (e.g., vertebral boundaries) and stabilizes gradient flow.

*Mamba Block.* Global dependencies are modeled efficiently through a three-step state-space formulation:

$$\mathbf{H} = \mathbf{W}_{\text{in}} \cdot \mathbf{F}_r, \tag{7}$$

$$\mathbf{Y} = \text{SelectiveScan}(\mathbf{H}; \mathbf{A}, \mathbf{B}, \mathbf{C}, \mathbf{D}), \tag{8}$$

$$\mathbf{F}_{\text{out}} = \mathbf{W}_{\text{out}} \cdot \mathbf{Y}, \tag{9}$$

where $\mathbf{W}_{\text{in}}, \mathbf{W}_{\text{out}}$ are learnable projection matrices, and $\mathbf{A}, \mathbf{B}, \mathbf{C}, \mathbf{D}$ are trainable state-space parameters. The selective scan operator enables linear-time sequence modeling, avoiding the quadratic cost of self-attention.

*SE Layer.* To highlight cardiac-relevant channels, the output is recalibrated via a squeeze-and-excitation mechanism. Global average pooling first aggregates context:

$$\mathbf{g} = \tfrac{1}{H \times W} \sum_{i=1}^{H} \sum_{j=1}^{W} \mathbf{F}_{\text{out}}, \tag{10}$$

where $\mathbf{g}$ is a channel descriptor. Two fully connected layers then rescale channels:

$$\mathbf{F}_{\text{se}} = \mathbf{F}_{\text{out}} \cdot \sigma(\mathbf{W}_2 \cdot \text{ReLU}(\mathbf{W}_1 \cdot \mathbf{g})), \tag{11}$$

with $\mathbf{W}_1, \mathbf{W}_2$ learnable matrices and $\sigma$ the sigmoid function.

**Regression Head.** The outputs of the four MambaStages (channels 64, 128, 256, 640) are processed sequentially, and the final-stage output is used via global average pooling and passed through a two-layer MLP with ReLU and a final linear layer to regress six cardiac key point coordinates. This head links hierarchical backbone features to precise anatomical localization, enabling reliable VHS computation. Together with the stem and MambaStages, it forms the full MambaVHS architecture, trained with the task-specific VHSAwareLoss to enhance accuracy and clinical consistency.

**VHSAwareLoss.** To stabilize VHS estimation, we introduce VHSAwareLoss, which combines regression, classification, and margin-based penalties with clinical thresholds (8.2, 10.0 VU). The loss consists of an L1 regression term, a classification penalty, and a soft margin term that reduces instability near decision boundaries. The $\delta$ term controls the boundary tolerance near the decision thresholds, while the middle multiplier $m$ adjusts the margin for cases near the 8.2-10.0 range, making the loss function more sensitive in this critical region. For further details, including full loss equations and derivations, please refer to Appendix A.2.

The base term is an L1 regression loss:

$$\mathcal{L}_{\text{reg}} = \left\| \widehat{VHS} - VHS \right\|_1, \tag{12}$$

augmented by a classification penalty

$$\mathcal{L}_{\text{cls}} = \mathbf{1}(\widehat{y} \neq y), \tag{13}$$

and a soft margin to reduce boundary instability:

$$\mathcal{L}_{\text{margin}} = \begin{cases} \text{ReLU}(\widehat{VHS} - (8.2 + \delta)), & y = 0, \\ \text{ReLU}(8.2 - \widehat{VHS}) + \text{ReLU}(\widehat{VHS} - (10 + \tfrac{\delta}{m})), & y = 1, \\ \text{ReLU}((10 - \delta) - \widehat{VHS}), & y = 2. \end{cases} \tag{14}$$

The final loss is:

$$\mathcal{L}_{\text{VHS}} = \mathcal{L}_{\text{reg}} + w_c(\mathcal{L}_{\text{cls}} + \mathcal{L}_{\text{margin}}). \tag{15}$$

## 5 EXPERIMENTS AND RESULTS

We evaluate MambaVHS on the CCK dataset through a series of experiments designed to measure both predictive accuracy and annotation reliability. First, we describe the training setup and compare MambaVHS against state-of-the-art baselines. We then analyze performance under L1 loss and conduct ablation studies to assess the contribution of individual components. Finally, we evaluate the VHSMarker annotation tool using Fleiss' Kappa to quantify inter-observer agreement.

## 5.1 MambaVHS Model

**Training Setup.** MambaVHS was trained with a joint objective of key point regression and classification to balance spatial accuracy and clinical relevance. We used the AdamW optimizer (learning rate $3 \times 10^{-4}$, weight decay $1 \times 10^{-6}$) with cosine annealing (minimum learning rate $1 \times 10^{-6}$). Gradient accumulation was applied to reduce memory cost, and checkpoints were selected by lowest validation loss. Training ran on a single NVIDIA A100 GPU with batch size 16, completing 150 epochs in about 22 hours. By comparison, other state-of-the-art models required $\sim$90 hours, highlighting the computational efficiency and rapid convergence of MambaVHS.

Table 3: Performance comparison of models trained with VHSAwareLoss on the CCK Dataset (test set). Accuracy, MSE, and MAE are reported, with MAE shown as mean $\pm$ standard deviation across multiple runs.

| Model | Accuracy (%) | MSE | MAE |
|---|---|---|---|
| GoogleNet | $78.75 \pm 0.30$ | $0.3741 \pm 0.015$ | $0.45921 \pm 0.41582$ |
| VGG16 | $78.00 \pm 0.28$ | $0.35287 \pm 0.014$ | $0.44912 \pm 0.37328$ |
| ResNet50 | $78.25 \pm 0.25$ | $0.31645 \pm 0.012$ | $0.43682 \pm 0.36417$ |
| DenseNet201 | $79.25 \pm 0.22$ | $0.34122 \pm 0.012$ | $0.42890 \pm 0.38674$ |
| InceptionV3 | $81.50 \pm 0.27$ | $0.26359 \pm 0.010$ | $0.37983 \pm 0.33921$ |
| Xception | $79.25 \pm 0.24$ | $0.31144 \pm 0.013$ | $0.41870 \pm 0.32964$ |
| Vision Transformer | $75.00 \pm 0.35$ | $0.47935 \pm 0.018$ | $0.47419 \pm 0.42367$ |
| ConvNeXt | $85.25 \pm 0.20$ | $0.19102 \pm 0.008$ | $0.34697 \pm 0.29911$ |
| EfficientNetB7 | $85.50 \pm 0.22$ | $0.28407 \pm 0.012$ | $0.38914 \pm 0.34973$ |
| CDA (Zhang et al., 2025) | $86.40 \pm 0.25$ | $0.21215 \pm 0.009$ | $0.35582 \pm 0.30763$ |
| MambaVision | $87.60 \pm 0.23$ | $0.20238 \pm 0.009$ | $0.33695 \pm 0.32479$ |
| **MambaVHS (Ours)** | $\mathbf{91.80 \pm 0.39}$ | $\mathbf{0.14380 \pm 0.015}$ | $\mathbf{0.212 \pm 0.1856}$ |

$^*$ $p < 0.05$ compared with all baselines (paired t-test, n = 4 runs).

**Model Evaluation.** This section presents the experimental evaluation of VHSMarker for vertebral heart score (VHS) estimation from canine thoracic radiographs. The primary evaluation metric is test accuracy, defined across three clinically meaningful categories: normal heart size ($< 8.2$), borderline cardiomegaly ($8.2 \leq \text{VHS} \leq 10$), and severe cardiomegaly ($> 10$).

Table 3 reports the performance of state-of-the-art baselines on the Canine Cardiac Keypoint (CCK) Dataset. In addition to accuracy, we also report mean squared error (MSE) and mean absolute error (MAE) to provide a more complete regression-based evaluation of keypoint localization and VHS estimation. The proposed MambaVHS model achieves the highest test accuracy of 91.8% ($\pm 0.39$), while also delivering the lowest MSE ($0.1438 \pm 0.015$) and MAE ($0.212 \pm 0.186$). These results highlight its strong capability in precise keypoint localization and clinically reliable VHS estimation. The margin of improvement over competitive baselines such as ConvNeXt (85.25%), EfficientNetB7 (81.50%), and CDA (86.4%) underscores the advantage of state-space modeling in capturing complex canine cardiac structures. Importantly, the CCK Dataset itself presents a challenging benchmark, as even advanced CNN and Transformer architectures plateau below 90% accuracy.

**MambaVHS Model Prediction Analysis.** Figure 4 compares VHS predictions from different models, including MambaVHS, ConvNeXt(Liu et al., 2022b), EfficientNetB7(Tan & Le, 2019), and CDA(Zhang et al., 2025), on canine thoracic radiographs. MambaVHS consistently generates predictions closer to the actual VHS, particularly for less common cases with irregular thoracic structures and unusual imaging angles. This highlights its superior ability to capture long and short axes accurately, outperforming other models in challenging scenarios, making it a reliable choice for real-world veterinary diagnostics.

**Ablation Study.** To assess the impact of architectural components and training strategies in MambaVHS, we performed a series of ablation experiments. These experiments systematically remove or replace specific modules to evaluate their contribution to overall performance.

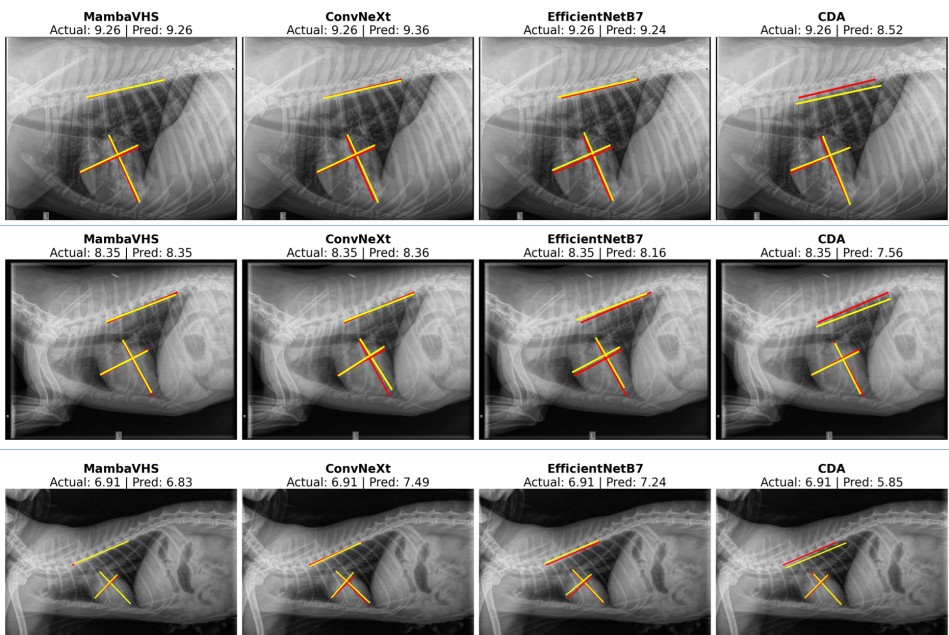

Figure 4: Comparison of VHS predictions for different deep learning models on canine thoracic radiographs. The ground truth is shown in Red, while predictions are shown in Yellow.

Table 4: Performance comparison of models trained

(a) With Component Ablations.

| Model Variant | Val Acc (%) | Test Acc (%) |
|---|---|---|
| Without SE Layers | 88.0 | 88.5 |
| With Attention + MLP | 80.1 | 84.7 |
| Without Residual Blocks | 82.0 | 84.5 |
| **Full Model** | **89.5** | **91.8** |

(b) With L1 loss.

| Model | Val Acc (%) | Test Acc (%) |
|---|---|---|
| MambaVision (Gu & Dao, 2023) | 86.55 | 87.60 |
| Swin Transformer (Liu et al., 2021) | 78.90 | 79.20 |
| ConvNeXt (Liu et al., 2022b) | 87.30 | 87.50 |
| CDA (Zhang et al., 2025) | 83.40 | 85.70 |
| EfficientNetB7 (Tan & Le, 2019) | 86.11 | 87.45 |
| MambaOut (Gu & Dao, 2023) | 83.45 | 85.78 |
| **MambaVHS (Ours)** | **88.40** | **89.70** |

Table 4a isolates the effect of individual design choices in MambaVHS: removing SE layers, removing residual blocks, or replacing the Mamba block with Attention+MLP consistently degrades performance, indicating each component is necessary for full accuracy. Moreover, under a fairness control where *all* models are trained with the same L1 regression loss (Table 4b), MambaVHS remains superior (88.40% / 89.70%), demonstrating that the gains stem from architecture rather than task-specific loss design.

**Agreement (Bland–Altman).** To further validate model reliability, we evaluate method–expert agreement on continuous VHS values using the Bland–Altman difference analysis (Bland & Altman, 1986; Giavarina, 2015). For each sample, the difference is defined as $d_i = \text{VHS}_i^{\text{model}} - \text{VHS}_i^{\text{expert}}$ and the mean as $m_i = (\text{VHS}_i^{\text{model}} + \text{VHS}_i^{\text{expert}})/2$. On the test set, the mean bias is $\bar{d} = +0.08$ VU with $\text{SD}_d = 0.28$ VU, producing 95% limits of agreement of $\bar{d} \pm 1.96\,\text{SD}_d = [-0.47, 0.63]$ VU. This narrow interval suggests that, across the clinical spectrum of cardiomegaly, MambaVHS predictions are consistently close to expert assessments, with deviations well within acceptable diagnostic tolerance reported in veterinary practice (Buchanan & Bücheler, 2000; Bélanger et al., 2014). These results indicate negligible systematic error and bounded dispersion, providing strong evidence that the model can serve as a reproducible adjunct to expert evaluation.

Confidence intervals are omitted here for brevity but can be provided in an extended version.

**External Validation with Task Specific Baseline Comparison.** To strengthen the generalization analysis, we evaluate MambaVHS on an external dataset of 2,000 canine thoracic radiographs collected independently by Shanghai Aichong Pet Hospital Li & Zhang (2024). No fine-tuning or hyperparameter changes are applied.

As summarized in Table 5, MambaVHS achieves 89.5% validation accuracy, 90.1% test accuracy, and an MAE of 0.23 VU, outperforming all previously reported CNN and Transformer baselines on this dataset. These results indicate strong cross-hospital transfer and anatomically grounded feature learning. The model maintains high reliability despite differences in equipment, acquisition protocols, and breed distributions, further supporting its robustness for multi-institution evaluation.

Table 5: External dataset performance.

| Model | Val Acc (%) | Test Acc (%) |
|---|---|---|
| GoogleNet | 78.0 | 75.8 |
| VGG16 | 79.0 | 75.5 |
| ResNet50 | 80.5 | 78.0 |
| DenseNet201 | 77.5 | 81.8 |
| InceptionV3 | 79.5 | 80.5 |
| Xception | 79.0 | 75.8 |
| InceptionResNetV2 | 78.0 | 79.5 |
| NasnetLarge | 79.5 | 83.8 |
| EfficientNetB7 | 82.5 | 85.5 |
| ViT | 79.5 | 79.5 |
| CONVT | 82.5 | 87.5 |
| Beit-Large | 71.5 | 75.0 |
| RVT | 85.0 | 87.5 |
| **MambaVHS** | **89.5** | **90.1** |

**Interpretability and Uncertainty.** We assess whether MambaVHS relies on anatomically meaningful cues using five saliency methods: Grad-CAM Selvaraju et al. (2017), Grad-CAM++ Chattopadhyay et al. (2018), Score-CAM Wang et al. (2020), Layer-CAM Jiang et al. (2021), and their ensemble. All methods consistently highlight cardiac borders and vertebral edges—the anatomical structures defining VHS (Fig. 5). To quantify stability, we compute inter-CAM agreement. Across 4,275 images, agreement is $0.83 \pm 0.05$, and the highest-uncertainty 10% of cases (low agreement) align with borderline VHS ranges where expert disagreement is also highest. These results indicate both anatomical plausibility and predictable uncertainty behavior. More details are provided in Appendix A.4.

## 5.2 VHSMARKER ANNOTATION TOOL

The VHSMarker tool was developed for efficient and accurate key point placement in canine thoracic radiographs. Its effectiveness is evaluated in terms of efficiency, usability, and annotation reliability. We also report the performance of the MambaVHS model trained on the CCK Dataset, which accurately predicts cardiac landmarks and estimates VHS in a fully automated manner. VHSMarker reduces annotation time to 10–12 seconds per image (vs. $\geq$1 min with MATLAB tools such as Li & Zhang (2024)); annotating 21,465 images required about 75 hours compared to 357 hours, a $4.8\times$ speedup.

Table 6: Fleiss' $\kappa$ by expert (n=9)

| Expert | Score | Expert | Score | Expert | Score | Expert | Score | Expert | Score |
|---|---|---|---|---|---|---|---|---|---|
| E1 | 0.81 | E2 | 0.81 | E3 | 0.85 | E4 | 0.86 | E5 | 0.89 |
| E6 | 0.90 | E7 | 0.91 | E8 | 0.93 | E9 | 0.94 | | |
| | | | | | | | | **Avg.** | **0.88** |

**Inter-observer Study.** To assess annotation consistency, we conducted an inter-observer study on 300 randomly sampled radiographs annotated independently by nine multidisciplinary experts. As shown in Table 6, Fleiss' $\kappa$ (Fleiss, 1971; McHugh, 2012) values ranged from 0.81 to 0.94, with an average of 0.88. According to the Landis–Koch scale (Landis & Koch, 1977), this corresponds to "almost perfect" agreement ($\kappa \geq 0.81$), confirming that VHSMarker enables efficient and highly reliable annotations across observers. A detailed comparison of annotation modes (Manual vs. Model-assisted vs. Hybrid) is provided in Appendix A.6, demonstrating that model-assisted workflows significantly reduce annotation time while maintaining annotation quality.

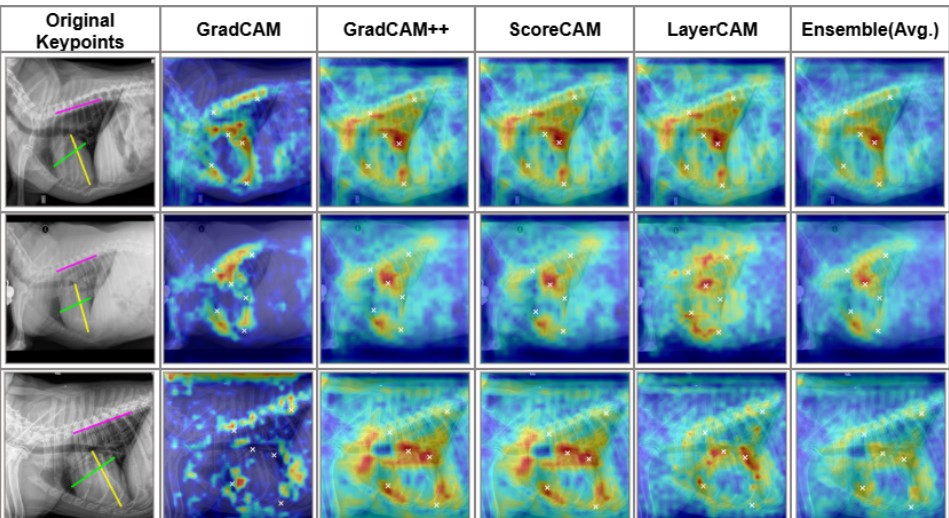

Figure 5: Representative CAM visualizations. All methods focus on cardiac borders and vertebral landmarks used in VHS measurement.

## 6 DISCUSSION

The VHSMarker framework reduces annotation time to 10–12 seconds per image through real-time feedback, responsive scaling, and intuitive interactions, lowering cognitive load and minimizing errors. The resulting CCK Dataset provides standardized annotations across diverse body sizes, anatomical variations, and clinical conditions, improving the reliability of downstream models such as MambaVHS. Together, the tool, dataset, and model form a scalable and precise pipeline for automated cardiomegaly assessment.

The CCK Dataset is currently limited to lateral thoracic views, has not yet been extended to other animal species, and shows an imbalanced breed distribution dominated by "Unknown" and mixed cases. This reflects real-world clinical records and does not directly affect VHS prediction, though stratified sampling could be explored in future work. While MambaVHS may face challenges with highly irregular anatomies or noisy images, such degraded radiographs are rarely used in clinical practice. Importantly, VHSMarker is designed for adaptability: small adjustments allow it to handle variations from different institutions or imaging devices, and species-specific VHS rules could be readily incorporated to extend the system beyond dogs. We plan to expand the tool to support collaborative annotation, including side-by-side comparison, exporting in standard formats (e.g., majority voting, STAPLE), and computing real-time inter-annotator agreement metrics. Future enhancements may further integrate self-supervised or active learning to reduce manual effort, or reinforcement learning to refine annotation efficiency and robustness. These directions highlight the flexibility of the framework and its potential as a foundation for scalable, clinically reliable AI systems.

## 7 CONCLUSION

In this work, we introduced VHSMarker, a fast and clinician-friendly annotation tool for canine thoracic radiographs, and used it to construct the large-scale CCK Dataset with over 21k standardized examples. Building on this resource, we proposed MambaVHS, a state-space based baseline model that achieves 91.8% test accuracy, outperforming strong CNN and Transformer counterparts. Together, these contributions establish the first unified benchmark for automated vertebral heart score estimation, reducing annotation time to under 10 seconds per image while improving predictive reliability. Beyond veterinary cardiology, this framework illustrates how efficient annotation pipelines combined with state-space architectures can enable scalable and clinically reliable AI systems, offering a foundation for broader applications in both animal and human healthcare.

## USE OF LARGE LANGUAGE MODELS

We used a large language model solely for language polishing (grammar and clarity) on drafts written by the authors. The LLM did not generate technical content, equations, code, analyses, figures, or results, and it was not used for ideation, literature search, data labeling, or experiments. All scientific claims and evaluations were produced and validated by the authors.

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

# A APPENDIX

## A.1 VHSMARKER ANNOTATION MODES

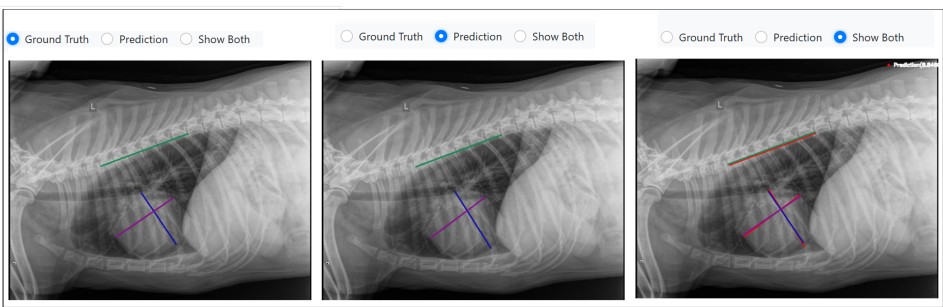

Figure 6: VHSMarker annotation modes: Ground Truth, Prediction, and Show Both, enabling precise adjustment and model comparison.

The three annotation modes are:

- **Ground Truth:** for manual labeling of cardiac key points.
- **Prediction:** for automated visualization of MambaVHS predictions.
- **Show Both:** for side-by-side comparison and adjustment.

These options streamline annotation, error correction, and model evaluation during large-scale dataset creation.

## A.2 VHS AWARE LOSS

---

**Algorithm 1** VHSAwareLoss Calculation

---

**Inputs:** Predicted VHS $v^{\text{pred}}$, True VHS $v^{\text{true}}$, thresholds $\tau_1{=}8.2$, $\tau_2{=}10$, margin $\delta$, middle multiplier $m$, class weight $w_c$
**Output:** $\mathcal{L}_{\text{VHS}}$
**1. Derive classes from thresholds**
$$y \leftarrow \begin{cases} 0 & \text{if } v^{\text{true}} < \tau_1 \\ 1 & \text{if } \tau_1 \leq v^{\text{true}} < \tau_2 \\ 2 & \text{if } v^{\text{true}} \geq \tau_2 \end{cases}$$
$$\hat{y} \leftarrow \begin{cases} 0 & \text{if } v^{\text{pred}} < \tau_1 \\ 1 & \text{if } \tau_1 \leq v^{\text{pred}} < \tau_2 \\ 2 & \text{if } v^{\text{pred}} \geq \tau_2 \end{cases}$$
**2. Base regression term (L1)**
$\mathcal{L}_{\text{reg}} \leftarrow |v^{\text{pred}} - v^{\text{true}}|$
**3. Class mismatch penalty**
$\mathcal{L}_{\text{cls}} \leftarrow \mathbb{1}[\hat{y} \neq y]$
**4. Margin-aware boundary penalty**
$$\mathcal{L}_{\text{margin}} \leftarrow \begin{cases} \max(0, v^{\text{pred}} - (\tau_1 + \delta)), & y = 0 \\ \max(0, \tau_1 - v^{\text{pred}}) + \max\left(0, v^{\text{pred}} - (\tau_2 + \frac{\delta}{m})\right), & y = 1 \\ \max(0, (\tau_2 - \delta) - v^{\text{pred}}), & y = 2 \end{cases}$$
**5. Final loss**
$\mathcal{L}_{\text{VHS}} \leftarrow \mathcal{L}_{\text{reg}} + w_c \left(\mathcal{L}_{\text{cls}} + \mathcal{L}_{\text{margin}}\right)$
**return** $\mathcal{L}_{\text{VHS}}$

---

The algorithm above defines the VHSAwareLoss calculation used for training the model in a task where VHS (Vertebral Heart Score) estimation is important. It takes as input the predicted VHS

$v^{\text{pred}}$, the true VHS $v^{\text{true}}$, predefined thresholds $\tau_1$ and $\tau_2$, a margin $\delta$, a middle multiplier $m$, and a class weight $w_c$. First, it determines the classes for both the true and predicted VHS values based on the thresholds: class 0 for values below $\tau_1$, class 1 for values between $\tau_1$ and $\tau_2$, and class 2 for values above $\tau_2$. The loss function consists of three components: (1) a base regression term (L1 loss), which penalizes the absolute difference between predicted and true VHS values, (2) a class mismatch penalty, which adds a loss when the predicted class does not match the true class, and (3) a margin-aware boundary penalty, which adjusts the loss based on the margin $\delta$ around the class boundaries. The final VHS loss is a weighted sum of these three terms, incorporating the class weight $w_c$ to balance the importance of class mismatch and margin penalties. This loss function encourages the model to predict VHS values that are not only close to the true value but also correctly classified within the specified boundaries.

## A.3 BREED INFORMATION

For completeness, Table 7 lists the full breed distribution of the CCK dataset, complementing the summary presented in Section 3.

Table 7: Complete breed distribution.

| # | Breed | Count |
|---|---|---|
| 1 | Mixed Dog | 1256 |
| 2 | Labrador Retriever | 479 |
| 3 | Golden Retriever | 191 |
| 4 | German Shepherd | 164 |
| 5 | Chihuahua | 100 |
| 6 | Boxer | 79 |
| 7 | Shih Tzu | 77 |
| 8 | Yorkshire Terrier | 77 |
| 9 | French Bulldog | 76 |
| 10 | English Bulldog | 72 |
| 11 | Canine, NOS | 67 |
| 12 | Miniature Poodle | 62 |
| 13 | Siberian Husky | 61 |
| 14 | Border Collie | 57 |
| 15 | Beagle Hound | 56 |
| 16 | Pomeranian | 52 |
| 17 | Cavalier King Charles Spaniel | 51 |
| 18 | Pug | 48 |
| 19 | Boston Terrier | 44 |
| 20 | Jack Russell Terrier | 44 |
| 21 | Maltese | 44 |
| 22 | Australian Shepherd | 42 |
| 23 | Shetland Sheepdog | 42 |
| 24 | Rottweiler | 41 |
| 25 | English Cocker Spaniel | 35 |
| 26 | Great Dane | 34 |
| 27 | Bernese Mountain Dog | 32 |
| 28 | Miniature Schnauzer | 32 |
| 29 | Cock-A-Poo | 31 |
| 30 | Standard Poodle | 31 |
| 31 | Havanese | 30 |
| 32 | Dachshund, NOS | 28 |
| 33 | Doberman Pinscher | 26 |
| 34 | Labradoodle | 26 |
| 35 | Great Pyrenees | 24 |
| 36 | Smooth Miniature Dachshund | 23 |
| 37 | English Setter | 21 |
| 38 | Australian Cattle Dog | 20 |
| 39 | Toy Poodle | 20 |
| 40 | Chinese Sharpei | 20 |

| # | Breed | Count |
|---|---|---|
| 41 | Bichon Frise | 19 |
| 42 | American Bulldog | 18 |
| 43 | Pembroke Welsh Corgi | 18 |
| 44 | West Highland Terrier | 18 |
| 45 | Rhodesian Ridgeback | 18 |
| 46 | English Springer Spaniel | 17 |
| 47 | American Staffordshire | 17 |
| 48 | Miniature Pinscher | 16 |
| 49 | Brittany Spaniel | 16 |
| 50 | Long-Haired Std Dachshund | 14 |
| 51 | German Short-Haired Pointer | 14 |
| 52 | Terrier, NOS | 14 |
| 53 | Basset Hound | 13 |
| 54 | Newfoundland | 13 |
| 55 | Bull Mastiff | 12 |
| 56 | Long-Haired Mini Dachshund | 12 |
| 57 | Bulldog, NOS | 12 |
| 58 | Belgian Malinois | 12 |
| 59 | Lhasa Apso | 11 |
| 60 | Greyhound | 11 |
| 61 | Bull Terrier | 10 |
| 62 | Irish Setter | 10 |
| 63 | Catahula Leopard Dog | 9 |
| 64 | Saint Bernard | 9 |
| 65 | Cocker Spaniel, NOS | 9 |
| 66 | Cairn Terrier | 9 |
| 67 | Rat Terrier | 9 |
| 68 | Irish Wolfhound | 9 |
| 69 | Collie, NOS | 9 |
| 70 | Cane Corso | 8 |
| 71 | Red Bone Hound | 8 |
| 72 | Samoyed | 7 |
| 73 | Chesapeake Bay Retriever | 7 |
| 74 | Vizsla | 7 |
| 75 | Smooth Standard Dachshund | 7 |
| 76 | American Pit Bull Terrier | 7 |
| 77 | Whippet | 7 |
| 78 | Akita | 6 |
| 79 | Leonberger | 6 |
| 80 | Schipperke | 6 |
| 81 | American Eskimo Dog | 6 |
| 82 | Mexican Hairless | 6 |
| 83 | Coonhound | 5 |
| 84 | English Mastiff | 5 |
| 85 | Silky Terrier | 5 |
| 86 | German Wire-Haired Pointer | 5 |
| 87 | Weimaraner | 5 |
| 88 | Papillon | 5 |
| 89 | Scottish Terrier | 5 |
| 90 | Staffordshire Bull Terrier | 5 |
| 91 | Mastiff, NOS | 5 |
| 92 | Hound, NOS | 5 |
| 93 | Keeshond | 5 |
| 94 | Giant Schnauzer | 4 |
| 95 | Airedale Terrier | 4 |
| 96 | Coton De Tulear | 4 |
| 97 | Swiss Mountain Dog | 4 |
| 98 | English Shepherd | 4 |
| 99 | Nova Scotia Duck Tolling Retriever | 4 |
| 100 | Saluki | 4 |

| # | Breed | Count |
|---|---|---|
| 101 | Italian Greyhound | 4 |
| 102 | Flat-Coated Retriever | 4 |
| 103 | Shiba Inu | 4 |
| 104 | Treeing Walker Coonhound | 4 |
| 105 | Bloodhound | 3 |
| 106 | Chinese Crested | 3 |
| 107 | American Foxhound | 3 |
| 108 | Tibetan Terrier | 3 |
| 109 | Neapolitan Mastiff | 3 |
| 110 | Australian Heeler | 2 |
| 111 | Spinone Italiano | 2 |
| 112 | Briard | 2 |
| 113 | Old English Sheepdog | 2 |
| 114 | Borzoi | 2 |
| 115 | Alaskan Malamute | 2 |
| 116 | Norwegian Elkhound | 2 |
| 117 | German Long-Haired Pointer | 2 |
| 118 | Affenpinscher | 2 |
| 119 | Peke-A-Poo | 2 |
| 120 | Anatolian Shepherd | 2 |
| 121 | Wirehaired Pointing Griffon | 2 |
| 122 | Toy Manchester Terrier | 2 |
| 123 | Clumber Spaniel | 2 |
| 124 | Standard Schnauzer | 2 |
| 125 | Irish Water Spaniel | 1 |
| 126 | Shiloh Shepherd | 1 |
| 127 | Cardigan Welsh Corgi | 1 |
| 128 | American Bully | 1 |
| 129 | Japanese Chin | 1 |
| 130 | English Coonhound | 1 |
| 131 | Border Terrier | 1 |
| 132 | Setter, NOS | 1 |
| 133 | Tibetan Spaniel | 1 |
| 134 | American Cocker Spaniel | 1 |
| 135 | Australian Terrier | 1 |
| 136 | Welsh Terrier | 1 |
| 137 | Norfolk Terrier | 1 |
| 138 | Dalmatian | 1 |
| 139 | Pharaoh Hound | 1 |
| 140 | Springer Spaniel | 1 |
| 141 | Silken Windsprite | 1 |
| 142 | Wirehaired Standard Dachshund | 1 |
| 143 | Retriever, NOS | 1 |
| 144 | Soft-Coated Wheaten Terrier | 1 |
| 145 | Maremma Sheepdog | 1 |
| 146 | Unknown | 8039 |
| | **Total** | 12385 |

## A.4 AGREEMENT ANALYSIS AND UNCERTAINTY ESTIMATION

We assess model–expert agreement using a Bland–Altman analysis. The mean bias between model and expert VHS is $+0.08$ VU, with 95% limits of agreement $[-0.47, 0.63]$ VU, fully within accepted clinical variability (Buchanan & Bücheler, 2000; Bélanger et al., 2014).

Uncertainty is quantified by measuring consistency across CAM methods (Fig. 5). Let the five normalized CAM maps be $\{M_t(x)\}_{t=1}^5$, each flattened to $m_t(x) \in \mathbb{R}^{HW}$. The agreement score is

$$A(x) = \frac{2}{T(T-1)} \sum_{s<t} m_s^\top m_t,$$

and the uncertainty is defined as $U(x) = 1 - A(x)$. Cases above the 90th percentile of $U(x)$ are flagged for clinician review.

Table 8: CAM Ensemble Agreement Statistics (N=4,275)

| Metric | Mean | Std Dev |
|---|---|---|
| Agreement Score $A(x)$ | 0.83 | 0.05 |
| Uncertainty $U(x)$ | 0.17 | 0.05 |
| Flagged Cases ($U > \tau_{90}$) | 428 (10%) | |

A high agreement score ($0.83 \pm 0.05$) indicates stable attention patterns across different interpretability methods, and a 10% flag rate provides a practical balance between automation and clinical oversight.

## A.5 CLINICAL WORKFLOW

The clinician-in-the-loop workflow integrates CAMs and uncertainty into the prediction pipeline:

- The model outputs keypoints, VHS, CAM visualizations, and an uncertainty score for each radiograph;
- High-confidence cases ($U(x) \leq \tau$) may be auto-reported;
- Low-confidence cases ($U(x) > \tau$) are routed to clinicians, together with CAMs and keypoint overlays for rapid verification.

This process adds negligible computational cost and preserves the interpretability of the system.

The combination of CAM explanations, high inter-method consistency, and explicit uncertainty scoring provides transparent model behavior suitable for safe deployment.

### A.5.1 CLINICAL DEPLOYMENT CONSIDERATIONS

Model accuracy aligns with clinical expectations. Our MAE of $0.21 \pm 0.19$ VU is smaller than typical inter-expert variability ($\pm0.3$–$0.5$ VU) (Bélanger et al., 2014). The Bland–Altman limits of agreement ($[-0.47, 0.63]$ VU) fall within the widely accepted $\pm0.5$ VU tolerance (Buchanan & Bücheler, 2000). Performance is strongest in clearly normal or clearly enlarged hearts, with higher uncertainty only near clinical boundaries where expert disagreement is also common.

Proposed deployment workflow:

- High-confidence cases ($\sim$70%): automated VHS reporting,
- Borderline / uncertain cases ($\sim$30%): clinician verification,
- Estimated workload reduction: $\sim$90% (10–12s vs. 60+s per image).

This workflow facilitates consistent measurement, reduces reader fatigue, and improves prioritization of urgent cases for clinical review.

## A.6 HUMAN-IN-THE-LOOP ANNOTATION WORKFLOW

The table below summarizes the performance of different annotation modes used in the CCK dataset. The Fully Manual mode represents traditional annotation, while the Model-Assisted mode involves initial predictions from the model, with experts refining the predictions. The Hybrid (Show Both) mode allows experts to view both model predictions and ground-truth annotations for refinement. We report the time per image, the

Table 9: Human-in-the-loop Annotation Mode Comparison

| Annotation Mode | Time (s) | Corrections Needed | Final Agreement ($\kappa$) | VHS MAE |
|---|---|---|---|---|
| Fully Manual | 62 ± 5 | N/A | 0.88 | 0.21 |
| Model-Assisted | 15 ± 3 | 2.1 ± 1.2 pts | 0.89 | 0.20 |
| Hybrid (Show Both) | 12 ± 2 | 1.3 ± 0.8 pts | 0.90 | 0.19 |

number of corrections needed (in points), final agreement ($\kappa$), and the mean absolute error (MAE) for VHS estimates across each mode. These results show that Hybrid mode significantly reduces annotation time while maintaining or improving accuracy and agreement.

