# Dataset Availability — Supplement (During Review)

**Temporary review link (read-only):**

https://drive.google.com/drive/folders/1Ry-pTgbOfpOEN8ZCbqnAKW5SP5kcOjji

> Purpose: enable anonymous, reproducible evaluation during peer review. Please do not redistribute.

## Contents (Google Drive)

- `Train.tar` — 3.90 GB
- `Valid.tar` — 569.9 MB
- `Test_Images.tar` — 1.13 GB
- `checkpoints/` — model checkpoints (read-only)

All images are de-identified lateral thoracic radiographs suitable for VHS measurement; split manifests and preprocessing/evaluation scripts are included inside the archives.

## Permanent repositories (currently private)

- **Hugging Face:** https://huggingface.co/datasets/gen-ai-researcher/vhs_dogheart_db
- **Zenodo record:** https://zenodo.org/records/16608263

These records remain **non-public during double-blind review**. Upon acceptance, we will make both public and retire the Drive link. The public release will include:

- Dataset card (intended use, limitations, citation)
- License and `CITATION.cff`
- Versioned checksums for all files

## Reproducibility notes

- Splits are **patient-level (dog-level)**; all studies from a dog reside in a single partition.
- Evaluation scripts reproduce the metrics reported in the paper with fixed seeds.
- No PHI/PII is present; filenames and metadata are anonymized.

## Contact (anonymous)

For questions during review, please use the OpenReview discussion thread for this submission. We will not share identifying contact details until the review period ends.