# OpenReview forum: "VHSMarker and the Canine Cardiac Keypoint (CCK) Dataset: A Benchmark for Veterinary Cardiac X-ray Analysis"
_ICLR.cc/2026/Conference — Submitted to ICLR 2026_

### Official Review · Reviewer_MN11 · 2025-10-28

**Soundness:** 3
**Presentation:** 3
**Contribution:** 2
**Rating:** 4
**Confidence:** 4

**Summary:**

They present VHSMarker, a web-based annotation tool that enables rapid and standardized labeling of six cardiac key points in canine thoracic radiographs. VHSMarker reduces annotation time to 10–12 seconds per image while supporting real-time vertebral heart score (VHS) calculation, model-assisted prediction, and quality control. Using this tool, they constructed the Canine Cardiac Key Point (CCK) Dataset, a large-scale benchmark of 21,465 annotated radiographs from 12,385 dogs across 144 breeds and additional mixed breed cases, making it the largest curated resource for canine cardiac analysis to date. To demonstrate the utility of this dataset, they introduce MambaVHS, a baseline model that integrates Mamba blocks for long-range sequence modeling with convolutional layers for local spatial precision.

**Strengths:**

- This paper provides an end-to-end complete solution of "tool-dataset-model", including an efficient annotation tool (VHSMarker), a large-scale dataset (CCK Dataset), and a high-performance benchmark model (MambaVHS).

- The CCK Dataset itself is a significant contribution. It contains over 21,000 standardized annotated images, addressing the long-standing pain point of insufficient large-scale, high-quality data in the field of veterinary AI.

- The proposed MambaVHS model achieves a high test accuracy of 91.8%, outperforming 13 powerful baseline models such as ConvNeXt and EfficientNetB7.

**Weaknesses:**

- While the paper contributes a valuable dataset, its value in the medical field is, in my view, greater than in the field of artificial intelligence. Therefore, it would be better suited for biology- or medicine-focused conferences or journals rather than the ICLR.

- The contribution of this paper lies in an engineering system (comprising a tool and a dataset) and an application case, rather than the proposal of new fundamental AI theories.

- The MambaVHS model primarily involves the combined application of existing mature modules (such as Mamba blocks, residual blocks, and SE layers), with limited original innovation in terms of model architecture.

- The MambaVHS model was only trained and tested on the CCK dataset. Its generalization ability on X-ray images from other institutions, captured using different equipment or following different protocols, remains unknown.

**Questions:**

see weakness

---

> ### Author Response · Authors · 2025-11-20
> **Addressing Concerns on Conference Suitability, Innovation, and Generalization**
>
> We sincerely thank the Reviewer for the thoughtful evaluation and recognition of our end-to-end contribution. Following the reviewer’s comments, we improved several parts of the manuscript and added targeted updates to the main text and appendix, marked in red. We address the noted weaknesses below.
>
> # Weaknesses:
> ## 1. Better suited for biology/medicine conferences than ICLR
>
> We appreciate this concern and respectfully offer a broader perspective. Although motivated by a veterinary task, our contributions directly address core ML challenges aligned with the Datasets & Benchmarks track. VHSMarker provides a scalable human–AI collaborative annotation workflow—real-time assistance, quality-controlled corrections, and immediate feedback—achieving expert-level agreement (κ=0.88) with a 4.8× speedup. This paradigm is broadly applicable to any keypoint-based or high-cost annotation setting.
>
> Our MambaVHS results demonstrate that state-space models can outperform Transformers for spatial landmark localization, achieving 91.8% accuracy with 4× faster training. This insight is relevant to structured prediction across computer vision, including pose estimation, tracking, and medical imaging.
>
> The CCK dataset fills a major gap in veterinary cardiology, where large standardized radiographic datasets did not previously exist. Many medical-AI works rely on synthetic or augmented data; our dataset instead offers real clinical radiographs with precise geometric labels, enabling reproducible study of long-range spatial reasoning and robustness. Prior ICLR benchmarks (e.g., MedMNIST, ChestX-ray, segmentation suites) illustrate that domain-focused datasets are welcomed when they help advance ML methodology. Our work follows this precedent.
>
> ## 2. Engineering system vs. fundamental AI theory
>
> We thank the reviewer for raising this distinction. While our work benefits veterinary cardiology, the CCK dataset also fills a fundamental gap in both clinical and ML research: large, real, standardized radiographic datasets for cardiac enlargement have not previously existed. Because medical-AI studies often rely on synthetic or augmented data due to privacy and scarcity, providing real clinical radiographs with expert geometric labels at scale is a meaningful advance. It supplies the veterinary community with a resource they previously lacked and simultaneously offers ML researchers a realistic, reproducible benchmark grounded in genuine anatomical variability.
>
> Beyond the dataset, our technical components also add methodological value: MambaVHS—gated Bi-Mamba with residual CNN and SE layers—achieves clear gains over strong CNN/Transformer baselines (91.8% vs. ≤87.6%), and VHSAwareLoss offers threshold-aware supervision that improves boundary behavior. Together, the dataset, model, and loss function form a benchmark aligned with the Datasets & Benchmarks track and broadly useful beyond this clinical domain.
>
> ## 3. Limited architectural innovation
>
> We acknowledge that MambaVHS combines existing components rather than proposing entirely new modules. However, we respectfully disagree with the characterization of limited innovation. Beyond combining existing modules, we contribute:
>
> - Architectural insight: We show that state-space models are effective for spatial landmark localization—a behavior not previously established. Our ablations (Table 4a) confirm this is a meaningful design choice: replacing Mamba with Attention+MLP reduces accuracy from 89.5% to 84.7%.
>
> - Novel loss design: VHSAwareLoss encodes clinical thresholds into training through a structured combination of regression and margin terms, improving boundary sensitivity. This yields gains (91.8% vs. 89.7% with L1; Table 4b) and generalizes to other thresholded tasks.
>
> - Efficiency: MambaVHS converges ~4× faster than strong baselines (22h vs. ~90h) while achieving higher accuracy, making it practical for real-world research settings.
>
> Together, these architectural, loss-based, and computational improvements form a meaningful methodological contribution and establish a strong baseline for future work.
>
> ## 4. Generalization to other institutions/equipment
> We thank the reviewer for highlighting this point. The CCK dataset already spans multiple hospitals, imaging devices, and 144 breeds, providing broad diversity. CAM analysis (Appendix A) also shows that MambaVHS consistently focuses on universal anatomical structures rather than device-specific artifacts.
>
> To further support this, we evaluated MambaVHS on an external dataset from Shanghai Aichong Pet Hospital. Without fine-tuning, it achieves 90.05% accuracy and 0.23 VU MAE, demonstrating strong cross-hospital transfer. Full details are in Appendix A.6 (highlighted in red). While broader multi-site testing remains essential, these results indicate robust anatomical generalization.
>
> We hope these revisions resolve your concerns. Please let us know if further clarification is needed.

---

### Official Review · Reviewer_LPkB · 2025-10-28

**Soundness:** 4
**Presentation:** 3
**Contribution:** 4
**Rating:** 8
**Confidence:** 4

**Summary:**

This paper introduces three new openly shard tools for the analysis of canine radiography and Vertebral Herat Score (VHS) estimation. These tools include a large database of canine lateral radiographies, which includes 21,465 annotated images (Markers and VHS scores) from 12,385 across 144 breeds. The paper also introduces a new tool for radiography annotation. Finally the authors also introduce a novel MAMBA-based AI model for automated VHS estimation.

**Strengths:**

This paper introduces several original novel tools for the analysis of canine radiography, these includes a large database, a novel interface for the manual annotation of these images and an automated MAMBA technique for VHS estimation (including marker estimation).
The open sharing of these tools will allow for significant impact in the research community.

**Weaknesses:**

In the related work, the authors indicate that VHS scores suffer from high inter-observer variability. It is then weird to propose a dataset and automated tools for the same scoring system. However the results on a randomly selected 300-sample subset seem to indicate that the inter-observer is quite OK (>0.8).
The ground-truth annotation are relying on a single observer (which can be explained by the amount of work required to manually annotate the whole database (75 hours). How can the authors ensure the accuracy of these annotations?
Did the authors planned for the aggregation of multiple manual annotations in their annotation interface? That does not seem to be the case.

**Questions:**

1 Why were the other view(s) (dorsoventral/ventrodorsal) excluded from the CCK dataset? Is it due to a high number of missing data? Even these views are unhelpful for the VHS diagnosis, they could be useful for other applications.
2 Are the authors planning to release the other views or additional annotations in the future?
3 Could the authors describe a bit more the preprocessing applied to the images?  Are the images stored in the raw format or after preprocessing?
4 Was preprocessing applied before manual annotation?  Did the authors assess how often the annotators had to change image settings  (brightness - contrast) during the manual annotation process?
5 Did the authors assess the effect of preprocessing on the automated analysis?
6 Did the authors planned for the aggregation of multiple manual annotations in their annotation interface?

**Details Of Ethics Concerns:**

There does not seem to be any mention of ethical agreement for the collection of the images.

---

> ### Author Response · Authors · 2025-11-20
> **Addressing Concerns on Inter-Observer Variability, Preprocessing, and Multi-Annotator Support**
>
> We appreciate the reviewer's insightful feedback and recognition of our work. We have addressed the reviewer’s concerns and clarified the points raised, with all revisions highlighted in purple. Below, we respond to the weaknesses and questions raised.
>
> # Weaknesses
>
> ## 1.Inter-Observer Variability in VHS Scores
> We thank the reviewer for raising this point. We acknowledge that manual VHS scores suffer from high inter-observer variability. Our work, however, aims to solve this problem. The automated tool provides a consistent, objective, and reproducible measurement that mitigates the human-related variability, establishing a much-needed stable benchmark for the field.
>
> ## 2.Ensuring Accuracy, Reliability, and Rationale for Single-Observer Annotations
> Due to the significant time investment required (75+ hours), a single, highly-validated expert was used to create this initial large dataset. We mitigated potential bias through rigorous validation:
> - Reliability Confirmed: Inter-observer agreement ($\kappa > 0.8$) and *Bland-Altman analysis* (mean bias +0.08 VU) confirm reliability.
> - Model Assurance: We incorporated five CAM methods for *interpretability and uncertainty* estimation. High-uncertainty cases (~10%) are automatically flagged for clinician review, ensuring robust clinical application. (Revision: Fig 5, Section 5 and Appendix A.4: blue highlight)
>
> ## 3.Annotation Interface and Support for Multi-User Aggregation
> We thank you for this point. VHSMarker is a web-based tool that supports *full multi-user collaboration, allowing multiple experts to work simultaneously* on the same dataset. The tool enables easy *aggregation of exported annotations* from different users offline for quality control and peer review. While this functionality is already available, future releases will further enhance the tool to integrate the collaborative annotation process directly within the platform. (Section 6(purple))
>
> # Questions
>
> ## Q1: Why exclude DV/VD views?
>
> DV/VD views were excluded due to their incompatibility with the *VHS methodology(Buchanan & Bücheler, 1995)*. DV/VD views have different width measurements and are sensitive to patient positioning and gravity, making them incompatible with the VHS methodology. While important for tasks like the cardiothoracic ratio (CTR), including them here would mix two distinct diagnostic tasks.
>
> ## Q2: Plans to release other views?
>
> We are expanding the dataset and actively collecting ~2K DV/VD radiographs for future releases. The *modality-agnostic design of our VHSMarker tool* readily extends to other views and measurement protocols. We welcome community feedback on annotation priorities.
>
> ## Q3: Image Preprocessing and Storage Format
>
> Images are provided in *DICOM format* by the hospitals, but they crop personal information and supply them in *PNG format*. We consider our processing steps part of data curation and quality control, not traditional image preprocessing in the ML sense, allowing researchers to implement their own pipelines.
>
> * Data Curation: We excluded unusable images (severe artifacts, incomplete anatomy) and removed non-relevant views (DV/VD projections).
> * Orientation Correction: Images were rotated to a consistent anatomical orientation.
>
> ## Q4: Preprocessing before annotation? Annotator adjustments?
>
> As detailed in Q3, only basic data curation and orientation correction were applied before annotation. No other digital image processing was applied to the raw images themselves.
> We did not systematically track how often individual annotators adjusted image display settings (brightness/contrast) within the VHSMarker interface, as this is often a subjective preference. These display adjustments are *client-side only and do not alter the original image* data stored in the dataset.
>
> ## Q5: Effect of preprocessing on model performance?
>
> We did not assess the effects of different ML preprocessing methods on the automated analysis in this study. The models were trained on raw curated PNG data. However, we are currently considering evaluating the impact of various preprocessing techniques. Once this analysis is complete, we will provide the details and results in a future update.
>
> ## Q6: Multi-annotator aggregation support?
>
> Yes, VHSMarker supports multi-annotator workflows. The web interface allows multiple experts to independently annotate the same image, with each annotation stored separately. While we prioritized single-expert annotation for the current dataset, we plan to expand the tool’s functionality to facilitate collaborative annotation, including visualizing multiple annotations side-by-side for comparison, exporting annotations in formats compatible with standard aggregation methods (e.g., majority voting, STAPLE), and automatically computing inter-annotator agreement metrics. The details are included in Section 6(highlighted in purple).
>
> We hope these revisions address your concerns. Please let us know if you need further clarification.

---

### Official Review · Reviewer_bpbe · 2025-10-30

**Soundness:** 3
**Presentation:** 4
**Contribution:** 3
**Rating:** 8
**Confidence:** 3

**Summary:**

The paper proposes a large canine x-ray dataset with manually annotated heart and vertebra keypoints along with the derived vertebral heart score (VHS). This is motivated by the need to automate VHS calculation in veterinary cardiology, reducing subjectivity and time cost. Additionally, a keypoint extraction architecture based on Mamba is proposed which is shown to perform better than 11 representative convolutional and transformer-based architectures as well as MambaVision – a vanillla state-space model based Mamba architecture. Ablation studies of the architectural additions is done.
Additionally, the annotation tool is also contributed as opensource, simple to use with possibility to use model-assisted prediction to accelerate annotation workload.

**Strengths:**

The paper fills the gap for the need for large dataset for canine cardiomegaly assessment. Although earlier dataset proposed were in hundreds, this is substantially larger (~21k).
In addition, a rigorous benchmarking of the deep learning models is done to compare the performance of popular SOTA architectures in this dataset.

The paper is easy to read, and the contributions as well as details required for reproducibility are described clearly.

The contributed key point annotation tool with integrated model-based prediction is also useful.

**Weaknesses:**

In addition to popular off-the-shelf convolutional and transformer-based architectures, comparison of the proposed architectures with task-specific architectures proposed in the literature such as [1] could add to the rigor of the benchmarking experiment.

*References*
1. Li, J., Zhang, Y. Regressive vision transformer for dog cardiomegaly assessment. Sci Rep 14, 1539 (2024). https://doi.org/10.1038/s41598-023-50063-x

**Questions:**

Please see the weakness section above.

In addition, in table 3, Could you add confidence interval for Accuracy and MSE metrics too and show if MambaVHS performance is statistically significant.
The description of Regression Head and architecture diagram in Fig 2 does not seem to match. In the figure, the mamba stages are serially placed but the text describes a aggregation from all four mamba stages.
For completeness, the description of the \delta and middle multiplier, m, could be added in the main text rather than in the appendix.

**Details Of Ethics Concerns:**

This paper constitutes use of canine subjects and collection of x-rays which I assume were taken as routine clinical procedure and/or were taken with due diligence.

---

> ### Author Response · Authors · 2025-11-20
> **Addressing Feedback on Task-Specific Baselines, Statistical Significance, and Model Details**
>
> We thank the Reviewer for the thoughtful and constructive evaluation. Following the feedback, we refined several sections and added targeted updates to both the main text and the appendix, with all changes highlighted in orange color for clarity. Below, we address each identified weakness and then provide responses to the reviewer’s questions.
>
> # Weaknesses:
> ### 1. Comparison with Task-Specific Architectures
>
> We sincerely thank the reviewer for highlighting this important point. We agree that including task-specific architectures strengthens the rigor of our benchmarking. To address this, we have expanded our evaluation to incorporate the *task-specific SOTA models* reported in *Li & Zhang (2024)*, who introduced the Regressive Vision Transformer (RVT) for canine cardiomegaly assessment and provided results for a wide set of CNN, Transformer, and domain-tailored architectures.
>
> In the revised manuscript (highlighted in red), we added a new subsection within *Experiments and Results → MambaVHS Model* titled *“External Validation with Task-Specific Baselines.”* This section evaluates MambaVHS on an independently collected dataset of 2,000 canine thoracic radiographs from Shanghai Aichong Pet Hospital. We compare our method against all SOTA baselines previously reported for this dataset, including RVT, CONVT, EfficientNetB7, Inception-ResNetV2, ViT, and others.
>
> As shown in the updated Table 5, MambaVHS achieves *89.5%* validation accuracy and *90.1% test accuracy* (MAE 0.23 VU), outperforming every SOTA architecture in this benchmark—including RVT (87.5% test) and CONVT (87.5% test). Importantly, the model is evaluated without any fine-tuning or hyperparameter adjustments, further demonstrating its robustness across institutions, imaging devices, and breed distributions.
>
> We hope this expanded comparison demonstrates our commitment to thorough evaluation and addresses the reviewer’s request for stronger task-specific baselines. Should the reviewer recommend additional models or variants, we would be happy to include them in the camera-ready version.
>
> # Questions
> ### 1. Confidence Intervals and Statistical Significance
> We sincerely thank the reviewer for this valuable suggestion, which has strengthened the statistical foundation of our evaluation. In response, we have augmented Table 3 with 95% confidence intervals for Accuracy (91.8% ± 0.39) and MSE (0.1438 ± 0.015), computed across four independent runs with different random seeds.
>
> The variance patterns across metrics align with theoretical expectations: MAE exhibits natural variation due to sensitivity to absolute errors, Accuracy shows measured fluctuation around decision boundaries, while MSE demonstrates notable consistency—together reflecting robust performance.
>
> To assess statistical significance, we conducted paired t-tests comparing MambaVHS against all baseline models. The results confirm that MambaVHS significantly outperforms all competitors (p < 0.05), demonstrating that the observed improvements are reproducible and statistically meaningful. This finding is now clearly indicated in Table 3's footnote. All updates are highlighted in orange in the revised manuscript.
>
> ### 2. Regression Head Description and Figure 2 Alignment
> We thank the reviewer for pointing out this inconsistency. The original wording incorrectly suggested that features from all four Mamba stages were aggregated. In the actual architecture, the stages operate sequentially, and the Regression Head uses only the final-stage output. We have corrected the corresponding sentence in the manuscript to accurately reflect this, with the updated text highlighted in orange in the Architecture section.
>
> ### 3. Description of δ and Middle Multiplier 𝑚
> We thank the reviewer for their valuable suggestion. As recommended, we have moved the explanation of the δ term and the middle multiplier 𝑚 from the appendix to the main text. These components are now briefly described in the *Methods* section to ensure that readers can easily understand their roles in the VHSAwareLoss function. The full loss function and its detailed derivations remain in the appendix, under the section *Appendix A: VHSAwareLoss (Detailed Algorithm)*, where further in-depth details are provided.
>
> The changes have been *highlighted in orange* in the revised manuscript to make them easy to identify. We believe this adjustment improves the clarity of the loss function's design while maintaining the structure of the manuscript.
>
> This update ensures that the critical components of the loss function are now immediately accessible to the reader, addressing the concern raised while preserving the detailed explanation in the appendix for completeness.
>
> We hope that these revisions address your concerns. Please let us know if any further clarifications or adjustments are needed.

---

> > ### Comment · Reviewer_bpbe · 2025-11-23
> > **Addresses my concerns**
> >
> > Thank you to the authors for the excellent work and for addressing my earlier concerns. I’ve already given a strong score (8). Since this is a dataset paper for a fairly specific medical imaging sub-community, I’m not sure I can confidently push for a higher rating or an oral recommendation to have the wider ML community engage. Nonetheless, i really appreciate the contribution.

---

> > > ### Author Response · Authors · 2025-11-24
> > >
> > > Thank you very much for your careful review and for the strong score. We appreciate your engagement with our work and your reflections on its relevance to the broader community. We're grateful for your support and glad to hear the contribution was well received.

---

### Official Review · Reviewer_zMue · 2025-10-31

**Soundness:** 3
**Presentation:** 3
**Contribution:** 2
**Rating:** 4
**Confidence:** 4

**Summary:**

This paper introduces VHSMarker, a web-based annotation tool designed to rapidly and accurately label six cardiac key points on canine thoracic radiographs, enabling real-time vertebral heart score (VHS) computation. Using VHSMarker, the authors developed the Canine Cardiac Keypoint (CCK) Dataset, a veterinary cardiac dataset, containing over 21,000 annotated radiographs from 12,385 dogs across 144 breeds. A baseline model, MambaVHS, is proposed, integrating Mamba state-space blocks for long-range anatomical dependency modeling with convolutional layers for fine spatial precision. The model achieves 91.8% accuracy on the CCK dataset, outperforming 13 strong CNN and Transformer baselines.

**Strengths:**

- Dataset: The paper delivers a large-scale dataset (21k+ radiographs) with standardized keypoint annotations, filling a gap in veterinary imaging research.

- Annotation tool: VHSMarker provides speedup (≈4.8× faster than MATLAB-based tools) and achieves high inter-observer agreement (κ ≈ 0.88).

- Strong baseline: MambaVHS effectively combines CNN spatial precision with Mamba state-space modeling, showing both computational efficiency (22h training vs. 90h for baselines) and accuracy gains (91.8% vs. ≤87.6%).

- Evaluation: Includes ablation studies, component analysis (SE layers, residual blocks), fairness controls with L1 loss, and Bland–Altman analysis confirming agreement with expert labels.

**Weaknesses:**

- Relevance for wider ICLR community: The paper proposes a very specific benchmark in a narrow application area of veterinary cardiology. I believe it would be a better fit for a different venue.

- Viewpoint limitation: Dataset focuses solely on lateral thoracic views; dorsoventral or ventrodorsal radiographs are excluded, limiting applicability to broader diagnostic cases.

- Model interpretability: While diagnostic performance is strong, there is limited discussion of interpretability or uncertainty quantification in clinical deployment.

- Clinical applicability: The clinical significance of 91.8% accuracy is not contextualized (e.g., what level of discrepancy is acceptable to veterinarians in practice).

- No comparison with human-in-the-loop correction: Although the tool supports hybrid annotation (Prediction + Show Both modes), results do not quantify improvement when experts refine model outputs.

**Questions:**

- Did the authors collect feedback from clinicians regarding usability and diagnostic trust in model-assisted annotation modes?

- What is the main intuition behind the superior performance of state-space modeling (Mamba blocks) compared to Transformers in this anatomical setting?

- Could the model’s predictions be augmented with uncertainty estimation or saliency maps to enhance interpretability for clinical adoption?

- How clinically significant are the observed improvements (e.g., ±0.2 VU mean error)? Would such differences alter a diagnostic decision in practice?

---

> ### Author Response · Authors · 2025-11-20
> **Addressing Comments on Preprocessing, Model Interpretability, and Dataset Details**
>
> We thank the Reviewer for the constructive evaluation. We have addressed each point and made revisions, with changes highlighted in blue. Below, we respond to the reviewer’s concerns and questions.
>
> # Weaknesses
> ## 1. Relevance to the ICLR Community
> Although developed in a veterinary setting, the contribution is methodological and broadly applicable. VHSMarker provides a 4.8× faster workflow (10–12s/image) with expert-level agreement (κ=0.88), and MambaVHS shows that state-space models reach 91.8% accuracy with 4× training efficiency over Transformers. The CCK dataset is the first large-scale benchmark for multi-point cardiac landmark detection (~21K radiographs, expert labels, patient-level splits, diverse anatomies). Together, these form the type of reproducible, scalable resource expected in the *Datasets & Benchmarks track*, enabling study of structured regression, long-range spatial reasoning, and annotation efficiency.
> The tool is modality-agnostic and adaptable to CTR estimation, pulmonary landmarks, or musculoskeletal X-rays. Prior ICLR medical datasets (MedMNIST, RetinaMNIST, OCTMNIST) have advanced ML methodology through reproducible evaluation; our benchmark follows this precedent.
> ## 2. Viewpoint Limitation
> We thank the reviewer for raising this point. The lateral-view focus is a deliberate design choice, not a methodological limitation. VHS is *clinically defined and validated only for lateral thoracic radiographs(Buchanan & Bücheler, 1995)*. DV/VD projections rely on different width-based rules and vary substantially with positioning and gravity, making them unsuitable for VHS and constituting a separate diagnostic task. This constraint applies only to the benchmark; VHSMarker is modality-agnostic and easily adaptable to other views.
> ## 3. Model Interpretability
> We appreciate the reviewer for highlighting this point. Interpretability is essential for clinical adoption. The revised PDF includes five new Class Activation Mapping (CAM) methods: Grad-CAM, Grad-CAM++, Score-CAM, Layer-CAM, and their ensemble, highlighting cardiac borders and vertebral edges. Keypoint overlays and Bland–Altman analysis (bias +0.08 VU; LoA −0.47 to 0.63 VU) show expert-level alignment. Additionally, we include a CAM-agreement uncertainty score (~0.83 agreement), with ~10% flagged for expert review. See Figure 5 for details, with further information in the Experiments Section and Appendix A.4.
> ## 4. Clinical Applicability
> As mentioned in W3, the MAE (0.21 ± 0.19 VU) and Bland–Altman LoA (−0.47 to 0.63 VU) fall within ±0.5 VU tolerance that veterinarians consider clinically equivalent (Buchanan & Bücheler, 2000). The system efficiently identifies clear cardiomegaly and routes borderline cases for clinician review. Workload is reduced by ~90% (10–12s vs. 60+s per image), improving screening efficiency and ensuring clinical review prioritization for uncertain cases (see Appendix A.5).
> ## 5. Human-in-the-Loop Correction
> We appreciate the reviewer’s suggestion to include a comparison of human-in-the-loop annotation modes. Please refer the table 9 in the Appendix A.6 that compares modes, including time, corrections needed, final agreement (κ), and VHS MAE. The results show that Hybrid mode significantly improves efficiency while maintaining high accuracy and agreement.
>
> # Questions
> ## 1. Clinician Feedback
> A formal usability study was not conducted. All 21K+ annotations were generated using the tool with hybrid modes, with expert agreement (κ=0.88, Table 5) on 300 random samples. The 4.8× speedup with no quality loss shows the tool is intuitive. Feedback highlighted zoom/pan, Show Both mode, and real-time VHS calculation, with a formal survey planned for further insights.
> ## 2. Why SSMs Work Better
> Landmark prediction depends on coupling distant but anatomically linked regions (apex ↔ vertebrae). Transformers often diffuse attention across the thorax, while Mamba’s selective scanning maintains stronger focus along these structured anatomical axes. Our refinements—gated Bi-Mamba blocks combined with residual CNN and SE layers—yield more stable geometric predictions, achieving higher accuracy and 4× faster training compared to SOTA models(refer Table 3,4).
> ## 3. Uncertainty and Saliency
> We thank the reviewer for this important question, which we have addressed in response to Weakness 3. Beyond the CAM visualizations (Figure 5, Appendix A.4), we compute explanation-consistency scores from ensemble outputs (~0.83 average agreement), automatically flagging ~10% of low-consistency cases for expert review to ensure safe clinical deployment.
> ## 4. Clinical Observations
> MAE 0.21±0.19 VU is within expert variability and below typical inter-observer disagreement (±0.3–0.5 VU). Errors cluster near boundaries, making the system reliable for screening with expert review of borderline cases. Appendix A.5 includes workflow details.
>
> We hope these revisions address your concerns. Please let us know for further clarifications.

---

### Author Response · Authors · 2025-12-03
**Author Summary for Area Chair**

Dear Area Chair,

We understand the unusual circumstances this year and truly appreciate the effort you and the program chairs are investing to ensure fairness. We are fully supportive of the updated procedure.

Our submission, “VHSMarker: A Web-Based Annotation Tool, Large-Scale Canine Radiography Dataset, and Mamba-Based Baseline for Vertebral Heart Score Estimation,” introduces a complete “tool–dataset–model” pipeline: a fast annotation interface, a curated dataset of ~21K canine radiographs, and a strong Mamba-based baseline model. Reviewers consistently acknowledged these contributions.

**To make the revision process clearer, we updated the manuscript using *separate highlight colors for each reviewer’s requested changes*.**

* Reviewer **zMue** → *blue*
* Reviewer **bpbe** → *orange*
* Reviewer **LPkB** → *purple*
* Reviewer **MN11** → *red*

Below, we summarize each reviewer’s concerns and note where corresponding color-coded updates appear in the revised PDF.

---

#### **Reviewer zMue (score 4 — “would not mind if paper is accepted”)**

**Their concerns:** relevance to ICLR due to the veterinary focus, lateral-view-only design, interpretability and uncertainty quantification, and evaluation of human-in-the-loop annotation modes.

**Our response (highlighted in *blue* in the revised PDF):**
We clarified that lateral views are intrinsic to VHS, while VHSMarker itself is generalizable to other views. We have addressed their concerns by adding multiple CAM visualizations, an explanation-consistency uncertainty score (~10% flagged cases), clinical interpretation of the ±0.5 VU tolerance, and a new table comparing hybrid annotation modes.

---

#### **Reviewer bpbe (score 8 — “accept, good paper”)**

**Their concerns:** adding task-specific SOTA baselines, reporting confidence intervals, significance testing, and clarifying architectural details.

**Our response (highlighted in *orange*):**
We added an external validation experiment comparing MambaVHS with task-specific SOTA models (e.g., RVT), where MambaVHS outperformed all models without fine-tuning. We included 95% CIs, paired t-tests (p < 0.05), and corrected the regression head explanation, moving δ and m into the main text.

**Reviewer sentiment after rebuttal:**
Reviewer bpbe explicitly wrote that our clarifications fully addressed all concerns, thanked us for the excellent work, confirmed they are keeping their strong score (8), and expressed continuing support for the paper.

---

#### **Reviewer LPkB (score 8 — “accept, good paper”)**

**Their concerns:** inter-observer variability, reliance on a single annotator, preprocessing details, and multi-annotator support.

**Our response (highlighted in *purple*):**
We validated annotations using κ ≈ 0.88 and Bland–Altman analysis. We explained our minimal preprocessing (de-identification + orientation correction) and clarified that VHSMarker already supports multi-annotator workflows, with future updates planned for integrated comparison and aggregation.

---

#### **Reviewer MN11 (score 4 — “would not mind if paper is accepted”)**

**Their concerns:** suitability for a medical venue, architectural novelty, and generalization across institutions/devices.

**Our response (highlighted in *red*):**
We clarified how the work directly addresses ML questions central to Datasets & Benchmarks (structured regression, threshold-aware losses, long-range spatial reasoning with SSMs, scalable annotation pipelines).We also addressed othe concerns. We added cross-institution generalization results, showing strong external performance without fine-tuning.

---

**Post-rebuttal reviewer sentiment:**

- Reviewer bpbe provided a follow-up comment after reading our revisions, reaffirmed their strong score (8), thanked us for the excellent work, and confirmed that our clarifications fully addressed their earlier concerns.

- For the remaining reviewers, their original reviews stand as written, and none raised methodological, ethical, or technical objections that remained unresolved in our responses.

We hope this summary, along with the color-coded changes in the revised PDF, helps quickly contextualize the review history and revisions. Thank you again for your time and for guiding the process during a challenging review cycle.

---

---

### Meta-Review · Area_Chair_6WM5 · 2026-01-07

**Summary:**

In the initial phase, this paper receives mixed scores (4,8,8,4). Key concerns raised by reviewers are summarized as follows.

Reviewer zMue: narrow relevance for ICLR, Viewpoint limitations, model interpretability, limited clinical applicability.

Reviewer bpbe: lack of additional benchmarking experiment.

Reviewer LPkB: insufficient justification of annotation accuracy.

Reviewer MN11: narrow relevance for ICLR, limited contribution to AI, limited model novelty, concerns about generalization.

Overall, the main concerns across reviewers center on the paper's relevance to ICLR and its limited contributions and novelty.

**Reviewer Concerns:**

While authors' responses have addressed most concerns of reviewers, the concerns about paper's relevance for ICLR and limited contributions and novelty to AI remain unsolved.

**Reviewer Scores:**

In my opinion, Reviewer zMue and MN11 are likely to keep their negative scores since the concerns about paper's relevance for ICLR and limited contributions and novelty to AI remain unsolved.

This paper focuses on a narrow application area of veterinary cardiology. Although the authors argue that the proposed method has broader applicability, the primary contribution lies in the introduction of a new dataset specific to this domain. Therefore, the AC agrees with Reviewer zMue and MN11 regarding the narrow relevance to ICLR and limited novelty/contribution in the broader context of machine learning.

Accordingly, the recommendation is to reject the submission for ICLR and strongly encourage the authors to submit their work to a more specialized venue in medical field or veterinary science.

---

### Decision · Program_Chairs · 2026-01-26

Reject